# Early menarche and childbirth accelerate aging-related outcomes and age-related diseases: Evidence for antagonistic pleiotropy in humans

Yifan Xiang[1], Vineeta Tanwar[1], Parminder Singh[1], Lizellen La Follette[1], Vikram Pratap Narayan[1,2], Pankaj Kapahi[1,3]*

[1]The Buck Institute for Research on Aging, Novato, United States; [2]Department of Biology and Chemistry, Embry-Riddle Aeronautical University, Prescott, United States; [3]Department of Urology, University of California, San Francisco, San Francisco, United States

## eLife Assessment

This **important** study uses Mendelian Randomization to provide evidence that early-life reproductive phenotypes (i.e., age at onset of menarche and age at first birth) have a significant impact on numerous health outcomes later in life. The empirical evidence provided by the authors supporting the antagonistic pleiotropy theory is **solid**. Theories of aging should be empirically tested, and this study provides a good first step in that direction.

*For correspondence:
pkapahi@buckinstitute.org

## Abstract

**Background:** Aging can be understood as a consequence of the declining force of natural selection with age. Consistent with this, the antagonistic pleiotropy theory of aging proposes that aging arises from trade-offs that favor early growth and reproduction. However, evidence supporting antagonistic pleiotropy in humans remains limited.

**Methods:** Mendelian randomization (MR) was applied to investigate the associations between the ages of menarche or first childbirth and age-related outcomes and diseases. Ingenuity Pathway Analysis was employed to explore gene-related aspects associated with significant single-nucleotide polymorphisms (SNPs) detected in MR analysis. The associations between the age of menarche, childbirth, and the number of childbirths with several age-related outcomes were validated in the UK Biobank by conducting regression analysis of nearly 200,000 subjects.

**Results:** Using MR, we demonstrated that later ages of menarche or first childbirth were genetically associated with longer parental lifespan, decreased frailty index, slower epigenetic aging, later menopause, and reduced facial aging. Moreover, later menarche or first childbirth was also genetically associated with a lower risk of several age-related diseases, including late-onset Alzheimer's disease, type 2 diabetes, heart disease, essential hypertension, and chronic obstructive pulmonary disease. We identified 158 significant SNPs that influenced age-related outcomes, some of which were involved in known longevity pathways, including insulin-like growth factor 1, growth hormone, AMP-activated protein kinase, and mTOR signaling. Our study also identified higher body mass index as a mediating factor in causing the increased risk of certain diseases, such as type 2 diabetes and heart failure, in women with early menarche or early pregnancy. We validated the associations between the age of menarche, childbirth, and the number of childbirths with several age-related outcomes in the UK Biobank by conducting regression analysis of nearly 200,000 subjects. Our

results demonstrated that menarche before the age of 11 and childbirth before 21 significantly accelerated the risk of several diseases and almost doubled the risk for diabetes, heart failure, and quadrupled the risk of obesity, supporting the antagonistic pleiotropy theory.

**Conclusions:** Our study highlights the complex relationship between genetic legacies and modern diseases, emphasizing the need for gender-sensitive healthcare strategies that consider the unique connections between female reproductive health and aging.

**Funding:** Hevolution Foundation (PK). National Institute of Health grant R01AG068288 and R01AG045835 (PK). Larry L. Hillblom Foundation (PK), Larry L. Hillblom Foundation (PS), Glenn Foundation (VN).

## Introduction

Given the decline in the force of natural selection with age, genes that influence aging and age-related diseases are likely to be selected for their influence on early-life events (*Medawer, 1952*). The theory of antagonistic pleiotropy suggests that natural selection involves inherent trade-offs, where genes that are advantageous for early survival and reproduction may have detrimental effects later in life, contributing to the aging process and the development of age-related diseases (*Medawer, 1952*; *Byars and Voskarides, 2020*; *Wang and Cong, 2021*). Though evidence for antagonistic pleiotropy has been observed in invertebrate models, evidence for causal relationships in mammals, especially humans, is largely lacking (*Kumar et al., 2016*). The timing of reproductive events, such as menarche and childbirth, has long been recognized as a crucial aspect of human life history and evolution. However, accumulating evidence suggests that these reproductive milestones may have far-reaching implications (*Grundy and Tomassini, 2005*; *Day et al., 2016*).

Considering reproductive events are partly regulated by genetic factors that can manifest the physiological outcome later in life, we hypothesized that earlier or later onset of menarche and first childbirth could reflect broader genetic influences on longevity and disease susceptibility, serving as proxies for biological aging processes (*Figures 1 and 2*). Mendelian randomization (MR) is a term that applies to the use of genetic variation to address causal questions about how modifiable exposures influence different outcomes (*Sanderson et al., 2022*). Two-sample and two-step MR analyses were adopted in our research to explore the genetic causal associations using the inverse variance weighted (IVW) model. We leveraged recent studies that have identified single-nucleotide polymorphisms (SNPs) associated with early menarche or birth in humans to examine their role in antagonistic pleiotropy. After preprocessing, there were 209 and 33 SNPs included in MR analyses for the two exposures, age at menarche (*Loh et al., 2018*) and age at first birth (*Mills et al., 2021*), respectively.

## Methods

All the MR research data is from the public genome-wide association studies (GWAS) databases of the IEU Open GWAS project (https://gwas.mrcieu.ac.uk/) and PubMed. The target analysis is based on public datasets. The population data is from the UK Biobank. No definite personal information was included. No additional ethical approval was required for our study.

### Exposure data

Two traits were included as female reproductive activity exposures, involving age at menarche (*Loh et al., 2018*) (GWAS ID: ebi-a-GCST90029036) and age at first birth (*Mills et al., 2021*) (GWAS ID: ebi-a-GCST90000048).

### Outcome data

The aging outcomes involved general aging, organ aging/disease, and organ cancers (*Supplementary file 1*). Frailty index (*Atkins et al., 2021*) (GWAS ID: ebi-a-GCST90020053), father's age at death (*Pilling et al., 2017*) (GWAS ID: ebi-a-GCST006700), mother's age at death (*Pilling et al., 2017*) (GWAS ID: ebi-a-GCST006699), and DNA methylation GrimAge acceleration (*McCartney et al., 2021*) (GWAS ID: ebi-a-GCST90014294) were included as general aging outcomes. Specific aging diseases and aging levels included age at menopause onset (*Loh et al., 2018*) (GWAS ID: ebi-a-GCST90029037), late-onset Alzheimer's disease (LOAD) (*Kunkle et al., 2019*) (PMID: 30820047),

**Figure 1.** Research pipeline. (**A, B**) All the traits for exposures and outcomes included in the analysis. Exposures include age at menarche and age at first birth. Outcomes include general aging, organ aging and diseases, and organ cancers. (**C**) The primary analyses for Mendelian randomization (MR) research, including instrumental variables selection, two-sample MR analysis, and sensitivity tests. (**D**) The advanced analyses include mediator analysis of body mass index (BMI) with two-step MR, colocalization analysis on single-nucleotide polymorphism (SNP)–SNP level, genetic correlation analysis based on LDSC, and gene target analysis based on RNA and protein expression analysis and Ingenuity Pathway Analysis. (**E**) The results are validated based on the UK Biobank with regression analysis. LOAD, late-onset Alzheimer's disease; CHF, chronic heart failure; CKD, chronic kidney disease; COPD, chronic obstructive pulmonary disease; GAD, gastrointestinal or abdominal disease.

osteoporosis (**Dönertaş et al., 2021**) (GWAS ID: ebi-a-GCST90038656), type 2 diabetes (**Sakaue et al., 2021**) (GWAS ID: ebi-a-GCST90018926), chronic heart failure (CHF) (**Sakaue et al., 2021**) (GWAS ID: ebi-a-GCST90018806), essential hypertension (GWAS ID: ukb-b-12493), facial aging (GWAS ID: ukb-b-2148), eye aging (**Ahadi et al., 2023**) (PMID:36975205), cirrhosis (GWAS ID: finn-b-CIRRHOSIS_BROAD), chronic kidney disease (**Wojcik et al., 2019**) (GWAS ID: ebi-a-GCST008026), early onset chronic obstructive pulmonary disease (COPD) (GWAS ID: finn-b-COPD_EARLY), and gastrointestinal or abdominal disease (GAD) (**Dönertaş et al., 2021**) (GWAS ID: ebi-a-GCST90038597). Organ cancers

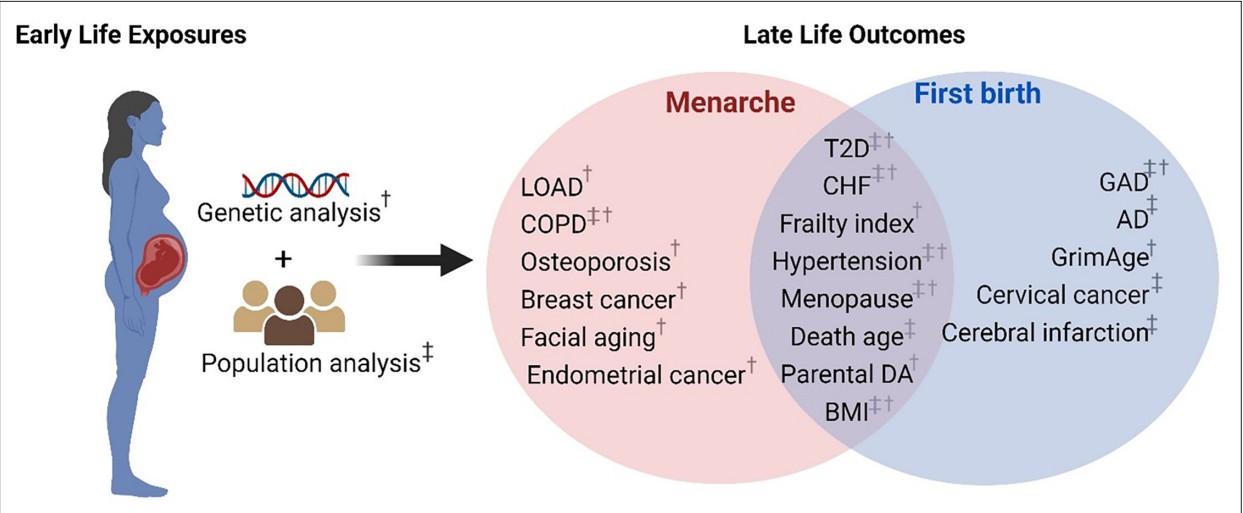

**Figure 2.** Schematic representation of early reproduction exposures and age-related outcomes. [†]Significant age-related outcomes in Mendelian randomization (MR) analysis. [‡]Significant age-related outcomes in UK Biobank (UKB) cohorts. LOAD, late-onset Alzheimer's disease; COPD, chronic obstructive pulmonary diseases; T2D, type 2 diabetes; CHF, chronic heart failure; BMI, body mass index; Parental DA, parental death age; GAD, gastrointestinal and abdominal diseases; AD, Alzheimer's disease. The figure is created with BioRender.com.

included breast cancer (*Sakaue et al., 2021*) (GWAS ID: ebi-a-GCST90018799), ovarian cancer (*Sakaue et al., 2021*) (GWAS ID: ebi-a-GCST90018888), endometrial cancer (*O'Mara et al., 2018*) (GWAS ID: ebi-a-GCST006464), and cervical cancer (GWAS ID: ukb-b-8777).

## SNPs selection

We identified SNPs associated with exposure datasets with $p < 5 \times 10^{-8}$ (*Savage et al., 2018*; *Gao et al., 2019*). In this case, 249 SNPs and 67 SNPs were selected as eligible instrumental variables (IVs) for exposures of age at menarche and age at first birth, respectively. All selected SNPs for every exposure would be clumped to avoid the linkage disequilibrium ($r^2 = 0.001$ and kb $= 10,000$). Then we identified whether there were potential confounders of IVs associated with the outcomes based on a database of human genotype–phenotype associations, PhenoScanner V2 (*Staley et al., 2016*; *Kamat et al., 2019*) (http://www.phenoscanner.medschl.cam.ac.uk/), with a threshold of $p < 1 \times 10^{-5}$. IVs associated with education, smoking, alcohol, activity, and other confounders related to outcomes would be excluded. During the harmonization process, we aligned the alleles to the human genome reference sequence and removed incompatible SNPs. Subsequent analyses were based on the merged exposure–outcome dataset. We calculated the *F* statistics to quantify the strength of IVs for each exposure with a threshold of $F > 10$ (*Burgess et al., 2017*). If the effect allele frequency (EAF) was missing in the primary dataset, EAF would be collected from dsSNP (https://www.ncbi.nlm.nih.gov/snp/) based on the population to calculate the F value. As osteoporosis GWAS was processed by BOLT-LMM, we recalculated the beta value for MR analysis. To adjust the beta values ($\beta_{BOLT}$) estimated by BOLT-LMM for dichotomous, we applied the following formula to each SNP:

$$Adjusted\ beta = \frac{\beta_{BOLT}}{\sqrt{P(1 - P)}}.$$

The prevalence (*P*) of the binary outcome in the population is the proportion of individuals with the outcomes. For example, the p-value for osteoporosis was calculated based on the GWAS sample size (484,598) and disease cases (7751). The adjusted beta values for essential hypertension, osteoporosis, GAD, and cervical cancer were calculated in MR analysis.

## MR analysis

All analysis was performed using R software (version 4.3.1, R Foundation for Statistical Computing, Vienna, Austria) and RStudio software (version 2023.09.0 Posit, PBC, Boston, USA) with TwoSampleMR (version 0.5.7) and MRPRESSO (version 1.0) packages.

Five two-sample MR methods were applied for analysis, including IVW model, MR Egger regression model, weighted median model (WMM), simple mode, and weighted mode. A pleiotropy test was used to check if the IVs influence the outcome through pathways other than the exposure of interest. A heterogeneity test was applied to ensure whether there is a variation in the causal effect estimates across different IVs. Significant heterogeneity test results indicate that some instruments are invalid or that the causal effect varies depending on the IVs used. MRPRESSO was applied to detect and correct potential outliers of IVs with NbDistribution = 10,000 and threshold p = 0.05. Outliers would be excluded for repeated analysis. The causal estimates were given as odds ratios (ORs) and 95% confidence intervals. A leave-one-out analysis was conducted to ensure the robustness of the results by sequentially excluding each IV and confirming the direction and statistical significance of the remaining SNPs.

In two-step MR, we mainly use IVW to assess the causal associations between exposure and outcome, exposure and mediator, and mediator and outcome. Body mass index (BMI) (GWAS ID: ukb-b-19953) was applied as a mediator. Similar steps of two-sample MR were repeated in the analysis after excluding confounders. The mediator effect (ME) and direct effect (DE) were calculated. The MEs were calculated with the formula: beta1 × beta2; DEs were calculated according to the formula: beta − (beta1 × beta2), beta stands for the total effect obtained from the primary analysis, beta1 stands for the effect of exposure on the mediator, and beta2 stands for the effect of the mediator on the outcome.

## Colocalization analysis

To improve the robustness of our research, colocalization analysis was conducted with packages gwasglue (version 0.0.0.9000), coloc (version 5.2.3), and gassocplot (version 1.0) between the exposures and outcomes revealing significant associations after BMI-related SNPs were excluded. SNPs with significant p-values in single SNP OR analysis were set as target SNPs. SNP-level colocalization was conducted with 50 kb (*Venkateswaran et al., 2018*) windows around each target SNP. EAF was set to 0.5 if it was missed in the primary datasets. The Bayesian algorithm in the coloc package generates posterior probabilities for the hypothesis that both traits are associated and share the same single causal variant at a specific lo (*Bulik-Sullivan et al., 2015a*; *Bulik-Sullivan et al., 2015b*) cus ($H_4$) (*Wallace, 2021*). SNPs with significant results in the colocalization analysis would be removed, and two-sample MR analyses would be conducted again.

## Genetic correlation analysis

Genetic correlation analysis helps clarify whether there is a shared genetic architecture between two traits, which can support the interpretation of our MR results. LDSC software, a robust framework to estimate genetic correlation using GWAS summary statistics, was applied to calculate the genetic correlation (*Bulik-Sullivan et al., 2015a*; *Bulik-Sullivan et al., 2015b*). GWAS summary statistics were filtered with HapMap3 variants, which were repeated or had a minor allele frequency ≤0.01 were excluded. Genetic correlation ($r_g$) was calculated for all pairwise comparisons among age at menarche, age at first birth, BMI, and age at menopause onset using LDSC method.

## Target analysis

We employed the Ingenuity Pathway Analysis (IPA) software (version 01-22-01; Ingenuity Systems; QIAGEN) to investigate various gene-related aspects associated with age at first birth and menarche. IPA is a widely used bioinformatics tool for interpreting high-throughput data (*Ghosh et al., 2017*). Briefly, the SNPs/genes from MR analysis were uploaded into QIAGEN's IPA system for core analysis and then the outcome was overlaid with the global molecular network in the Ingenuity Pathway Knowledge Base (IPKB). IPA was performed to identify canonical pathways, diseases, and functions, and to investigate gene networks. Additionally, the Chemical Biology Database (ChEMBL) (https://www.ebi.ac.uk/chembl/), a large-scale database of bioactive molecules for drug discovery (*Gaulton et al., 2012*), and the DrugBank database (*Wishart et al., 2018*), a comprehensive database on

FDA-approved drugs, drug targets, mechanisms of action, and interactions (https://go.drugbank.com/), were used to identify candidate genes targeted by approved drugs in clinical trials or under current development phases.

## Population validation

The MR results were further validated based on the UK Biobank with package stats (version 4.3.1). In addition to age at menarche and first live birth, the number of births is listed as an independent variable for the validation analysis. Based on the significant associations, regression analysis was adopted to explore the effect of independent variables and related confounders on outcomes. Ages at menarche were divided into five age groups, <11, 11–12, 13–14, 15–16, and >16 years. Ages at first live birth were divided into five age groups, <21, 21–25, 26–30, 31–35, and >35 years. Females with the number of births from 0 to 4 were included. Education was divided into seven levels based on the degrees. Smoking was divided into two categories, ever smoking or not. Drinking was divided into three categories: never, previous, and current drinking. BMI was divided into four categories: <18.5, 18.5–24.9, 25–29.9, and ≥30 based on the average value of four records. The outcome results were preprocessed based on the baseline information collection, follow-up outcome, surgical history, and ICD-10 diagnosis summary. Logistic regression was used for the analysis of disease risks (categorical variables). The coefficients ($\beta$) in logistic regression were log-odds ratios. Although the continuous variables exhibited mild to moderate skewed distributions, considering the large sample size, linear regression was used for the analysis of age at death, menopause age, and BMI in the first step. Then logistic regression was applied to confirm the robustness of the results with the outcome of death age ≥80, menopause age ≥50, and BMI ≥30. Confounders for regression analysis I include education, smoking, and drinking. Confounders for regression analysis II include education, smoking, drinking, and BMI. Variance Inflation Factors (VIF) were calculated for each predictor in regression analysis to avoid multicollinearity. Values of VIF were less than 1.5 for each predictor. For outcomes of diabetes, high blood pressure (HBP), heart failure, and BMI ≥30, the combined effect of age at menarche and first birth was analyzed. 179,821 participants were divided into 25 groups according to the menarche and first live birth age groups. All other groups were compared to the group with menarche of '13–14 years' and the first birth of '26–30 years' in logistic regression (confounders including education, smoking, drinking, and BMI for diabetes, HBP, and heart failure; including education, smoking, and drinking for BMI ≥30).

## Results

### Later menarche was genetically associated with later aging outcomes

Since parental ages at death may reflect the genetic heritability of lifespan, parental ages at death were used as general aging outcomes (*Ng and Schooling, 2023*). Compared to early menarche, later age at menarche was significantly associated with later parental ages at death for mothers or fathers (Beta = 2.27E−02, p = 1.84 × 10$^{-4}$; Beta = 1.97E−02, p = 7.88 × 10$^{-4}$ for father and mother's ages at death, respectively) (*Figure 3A* and *Supplementary files 1–3*). Aging is the biggest risk factor for frailty and several age-related diseases. Later age at menarche was associated with a lower frailty index, which was calculated using a set of 49 self-reported questionnaire items on traits covering health, presence of diseases and disabilities, and mental well-being (*Williams et al., 2019*) (Beta = −2.36E−02, p = 5.75 × 10$^{-3}$). After excluding the IVs associated with BMI, the significant association between later menarche and outcomes remained based on the WMM for frailty index (Beta = −2.88E−02, p = 0.0202) and based on IVW for parental ages at death (Beta = 1.47E−02, p = 2.46 × 10$^{-2}$; Beta = 1.24E−02, p = 3.87 × 10$^{-2}$). Next, we examined the associations between the age of menarche and organ aging or age-related diseases (*Figure 3A, B* and *Supplementary file 3*). Later age of menarche was associated with later menopause onset (Beta = 1.16E−01, p = 1.93 × 10$^{-2}$), lower risks of LOAD (OR = 0.897 (0.810–0.994), p = 3.88 × 10$^{-2}$), CHF (OR = 0.953 (0.913–0.995), p = 2.80×10$^{-2}$), essential hypertension (OR = 0.966 (0.959–0.973), p = 1.20 × 10$^{-22}$), facial aging (Beta = −1.76E−02, p = 2.16 × 10$^{-9}$), and early onset COPD (OR = 0.838 (0.755–0.931), p = 9.50 × 10$^{-4}$) and mildly higher risk of osteoporosis (OR = 1.009 (1.006–1.013), p = 1.95 × 10$^{-7}$). These associations were still significant after excluding SNPs associated with BMI, except for CHF. One potential mechanism of antagonistic pleiotropy is accelerated cell growth and division; therefore, we examined the

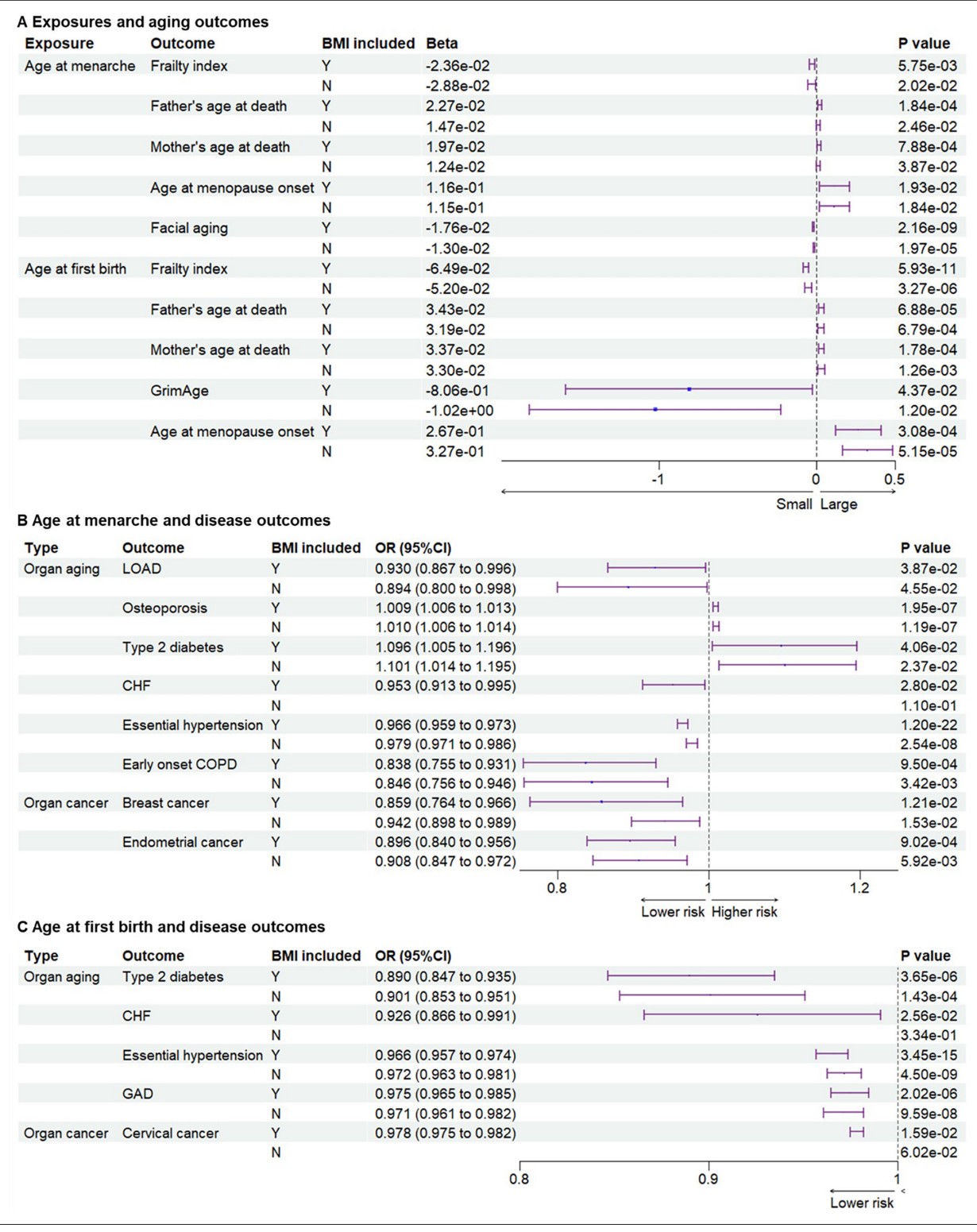

**Figure 3.** Genetic associations between exposures and outcomes. (**A**) Genetic associations between exposures and aging outcomes. Later age at menarche was associated with a lower frailty index and facial aging, higher parental ages at death, and later menopause. Later age at first birth was associated with lower frailty index and GrimAge, higher parental ages at death, and later menopause. (**B**) Genetic associations between age at menarche and outcomes of organ diseases. Later age at menarche was associated with lower risks of LOAD, CHF, essential hypertension, early onset COPD, breast cancer, and endometrial cancer. (**C**) Genetic associations between age at first birth and outcomes of organ diseases. Later age at first birth was associated with lower risks of type 2 diabetes, CHF, essential hypertension, gastrointestinal or abdominal disease, and cervical cancer. The

*Figure 3 continued on next page*

*Figure 3 continued*

significant associations were not detected after BMI-related SNPs were excluded for outcomes of CHF and cervical cancer. BMI included, BMI-related SNPs included; LOAD, late-onset Alzheimer's disease; CHF, chronic heart failure; COPD, chronic obstructive pulmonary; GAD, gastrointestinal or abdominal disease; BMI, body mass index.

relationship between menarche and certain cancers. Later age at menarche was associated with lower risks of breast cancer (OR = 0.859 (0.764–0.966), p = 1.53 × 10$^{-2}$) and endometrial cancer (OR = 0.896 (0.840–0.956), p = 9.02 × 10$^{-4}$) compared with early menarche (*Figure 3B*).

## Later age at first birth was genetically associated with later aging outcomes

Next, we examined the associations between the age of childbirth and age-related outcomes. Compared to early first birth, later age at first birth was associated with lower frailty index (Beta = −6.48E−02, p = 5.93 × 10$^{-11}$) and GrimAge, a measure of epigenetic aging (Beta = 2.67E−01, p = 4.37 × 10$^{-2}$), and older parental ages at death (Beta = 3.43E−02, p = 6.88 × 10$^{-5}$ for father's age at death; Beta = 3.37E−02, p = 1.78 × 10$^{-4}$ for mother's age at death) (*Figure 3A*). Similar to results with the age of menarche, later age at first birth was associated with later menopause onset (Beta = 2.67E−01, p = 3.08 × 10$^{-4}$), lower risks of type 2 diabetes (OR = 0.890 (0.847–0.935), p = 3.65 × 10$^{-6}$), CHF (OR = 0.926 (0.866–0.991), p = 2.56 × 10$^{-2}$), essential hypertension (OR = 0.966 (0.957–0.974), p = 3.45 × 10$^{-15}$), and GAD (OR = 0.975 (0.965–0.985), p = 2.02 × 10$^{-6}$). Most significant associations remained after the BMI-related SNPs were excluded, except for CHF (*Figure 3A, C* and *Supplementary file 3*). Furthermore, later age at first birth was also significantly associated with a lower risk of cervical cancer (OR = 0.978 (0.975–0.982), p = 8.67 × 10$^{-38}$) but not breast and endometrial cancers (*Figure 3C*).

## BMI is an important mediator in significant associations

As BMI is an important modulator of aging (*Locke et al., 2015*), we examined its role in explaining these associations. Based on the two-sample MR analyses, significant associations between early menarche and CHF, and between early first birth and CHF and cervical cancer, were not observed after excluding SNPs related to BMI (*Figure 3*). To estimate the effect of BMI as the mediator, we further conducted two-step MR. Exposures of age of menarche and age at first birth, and outcomes of frailty index, type 2 diabetes, and CHF, were included in the two-step MR analysis. Later ages of menarche and first birth were associated with lower BMI (Beta = −5.36E−02, p = 2.73 × 10$^{-13}$; Beta = −4.49E−02, p = 6.85 × 10$^{-5}$). Higher BMI was associated with higher risks of type 2 diabetes (OR = 2.391 (2.243–2.550), p = 4.65 × 10$^{-156}$), CHF (OR = 1.690 (1.570–1.819), p = 1.67 × 10$^{-44}$), and cervical cancer (OR = 1.019 (1.005–1.033), p = 8.18 × 10$^{-3}$). Significant pleiotropy was detected in the association between BMI and frailty index. Two-step MR showed that BMI significantly mediated the associations between menarche and type 2 diabetes or CHF (*Supplementary file 4*). BMI also significantly mediated the associations between first birth and type 2 diabetes, CHF, or cervical cancer (*Supplementary file 4*). As both early menarche and first birth were associated with higher BMI, BMI had a significant effect to explain the associations partially. Results of the colocalization analysis showed five SNPs associated with both the exposures and outcomes (*Supplementary file 5*). The genetic correlation results analyzed with the LDSC method are listed in *Supplementary file 6*.

## Mechanisms explaining antagonistic pleiotropy

We identified 158 significant SNPs in the genes from the MR analysis for their association between age of menarche or first birth with 15 aging and age-related disease outcomes. These SNPs were then listed according to (1) the number of study outcomes they were associated with and (2) their frequency of occurrence in the European and global population. We chose to depict only the SNPs that were associated with greater than three outcomes as shown in *Figure 4*. The SNP rs2003476 in the *CRTC1* gene was found to be associated with seven aging and age-related disease outcomes (frailty index, LOAD, CHF, father's age at death, mother's age at death, breast, and endometrial cancer). Previous studies have indicated that CRTC1 transcription domain is linked with extending the lifespan in *C. elegans* (*Silva-García et al., 2023*) and a decrease in CRTC1 levels was found to be associated with human Alzheimer's disease (AD) (*Mendioroz et al., 2016*). Two other genes involved in the glutathione metabolism pathway, *CHAC1* and *GGT7*, were also found to be associated with four different aging

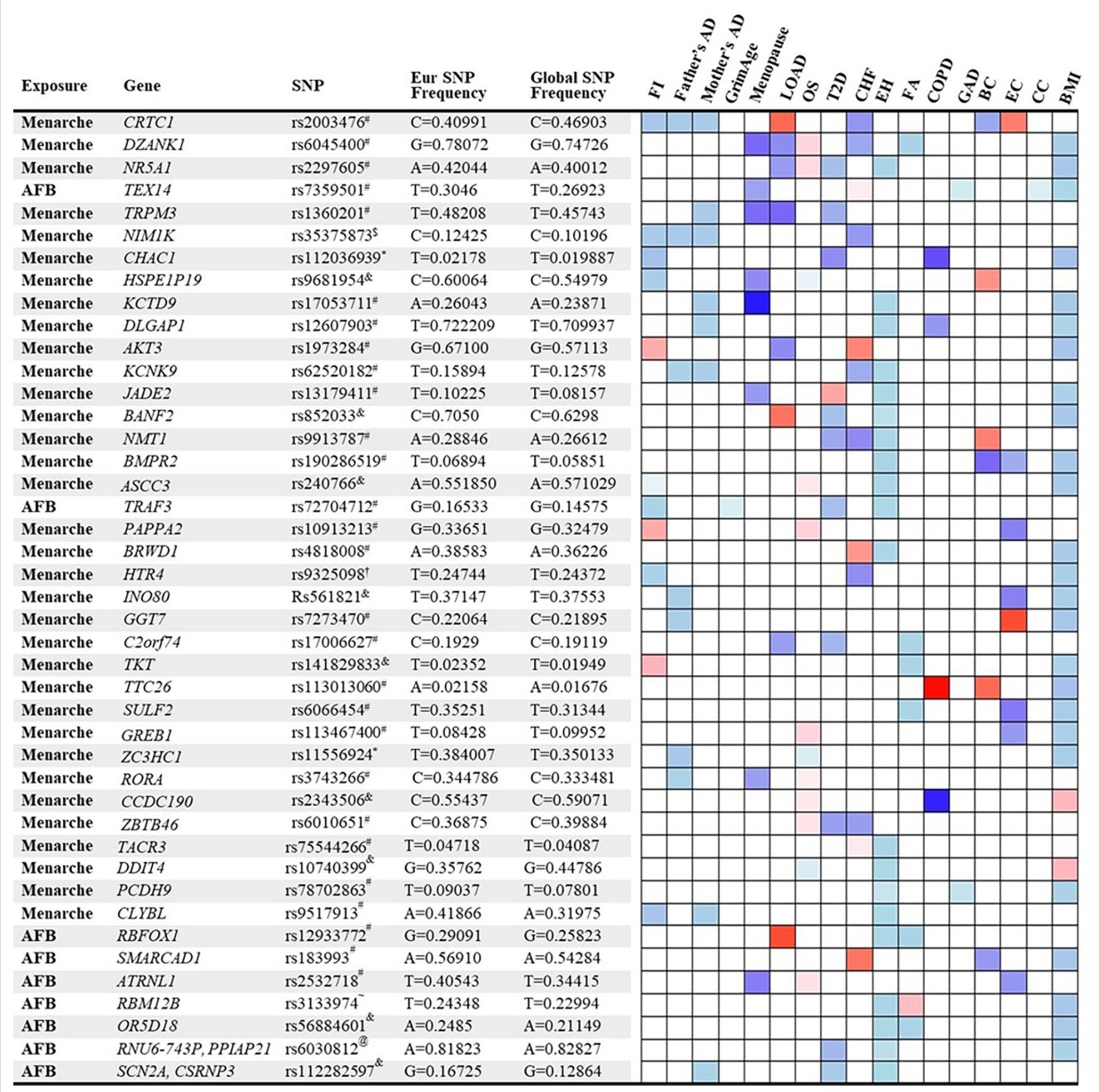

**Figure 4.** List and heatmap of single-nucleotide polymorphisms (SNPs)/genes from post-Mendelian randomization (MR) analysis showing an association between age at menarche and first birth with three or more aging outcomes. Genomic region location for each variant is depicted by different characters: intron variant#; 2 KB upstream variant$; Missense variant*; Intergenic variant&; Regulatory region variant@; 3' UTR variant†; Synonymous Variant~. A heatmap is representative of an association of each gene/SNP with different aging outcomes. Red bar represents a harmful association (−beta for Father's DA, Mother's DA, and menopause;+beta for other outcomes); The blue bar represents a beneficial association (+beta for Father's DA, Mother's DA, and menopause; −beta for other outcomes); the white bar represents no association. AFB, age at first birth; FI, frailty index; Father's DA, father's age at death; Mother's DA, mother's age at death; Menopause, age at menopause; LOAD, late-onset Alzheimer's disease; OS, osteoporosis; T2D, type 2 diabetes; CHF, chronic heart failure; EH, essential hypertension; FA, facial aging; COPD, chronic obstructive pulmonary disease; GAD, gastrointestinal or abdominal disease; BC, breast cancer; EC, endometrial cancer; CC, cervical cancer; BMI, body mass index.

outcomes (*CHAC1* was associated with frailty index, diabetes, COPD, and BMI; *GGT7* was associated with father's age at death, endometrial cancer, and BMI) (*Figure 4*). Importantly, the SNP in *CHAC1* is a missense variant, suggesting its impact on protein translation and potential contribution to disease biology. CHAC1 has recently been implicated in age-related macular degeneration (*Liu et al., 2023*) and muscle wasting (*Li et al., 2023*). Posttranslational protein–lipid modification processes are known to contribute to aging and age-related diseases (*Cloos and Christgau, 2004*). *N*-Myristoylation is one

such process that is catalyzed by NMT (*N*-myristoyltransferase) (*Udenwobele et al., 2017*), another intriguing gene candidate identified in our analysis. We found an association of *NMT1* with four aging outcomes: type 2 diabetes, CHF, essential hypertension, and breast cancer. Various myristoylated proteins have been implicated in diverse intracellular signaling pathways (*Hayashi and Titani, 2010*) including AMP-activated protein kinase (AMPK) (*Wen et al., 2019*; *Liang et al., 2015*). It has been shown that NMT1 exerts synovial tissue-protective functions by promoting the recruitment of AMPK to lysosomes and inhibiting mTORC1 signaling (*Wen et al., 2019*). In addition, drugs targeting NMTs (NMT1 and NMT2) have been suggested to be potent senolytics (*McHugh et al., 2023*) and a target for treating or preventing various diseases such as cancer (*Thinon et al., 2016*; *Zhu et al., 2021*) and heart failure (*Tomita et al., 2023*). Furthermore, modulating the evolutionarily conserved *N*-myristoyl transferase NMT1 was shown to extend lifespan in yeast (*Ashrafi et al., 2000*). The full table of SNPs and their population frequency and association with aging outcomes is provided in *Supplementary file 7*.

## Canonical signaling pathways analysis

A total of 563 canonical signaling pathways related to 158 SNPs/gene outcomes were identified based on the IPKB. After ranking the identified canonical signaling pathways according to their adjusted p-values, the top 25 canonical signaling pathways with a p-value $<10^{-2}$ enriched by SNPs/ genes involved in the age of menarche and first birth are represented in *Figure 5A*. The top enriched canonical signaling pathways fell into these broader categories: (1) Developmental and cellular signaling pathways, such as bone morphogenetic protein (BMP), insulin-like growth factor 1 (IGF-1), and growth hormone signaling, and signaling by Activin; (2) Neuronal signaling and neurological disorders, including glioma signaling, amyloid processing, activation of NMDA receptors and postsynaptic events, and neuropathic pain signaling; (3) Metabolic and endocrine signaling pathways, such as mTOR signaling, leptin signaling, PFKFB4 signaling, and melatonin signaling; (4) Immunological and inflammatory pathways, which include lymphotoxin β receptor and IL-22 signaling; (5) Cardiovascular and muscular signaling pathways, including the role of NFAT in cardiac hypertrophy. In addition to canonical pathways, SNPs/genes were shown to be further associated with diseases and functions. IPA showed a total of 78 diseases and functions associated with age at menarche and first birth. These associated diseases and functions were rated according to their adjusted p-values, and the top 20 enriched categories with a p-value $<10^{-5}$ were selected and depicted in *Figure 5B*.

## Gene network analysis and drug interactions

Besides pathway and disease associations, the 158 SNPs/genes were further connected to identify 12 functional networks using IPA. These IPA networks were ranked based on their consistency scores. The top 5 networks included a range of 13–22 genes with scores above 25, indicating robust regulatory analysis (*Krämer et al., 2014*; *Supplementary file 8*). The top networks were mainly connected to the following functions: cancer, hematological disease, immunological disease (*Figure 5C*), cardiovascular disease, organismal injury and abnormalities, reproductive system development and function (*Figure 5D*), and cell-to-cell signaling and interaction. These results support the notion that dynamic cellular interactions and the development of various vital organ systems influenced by age at menarche and first birth also influence age-related diseases. The two top IPA networks were further outlined in *Figure 5C, D*. To further explore the drugs targeting genes associated with age at menarche and first birth, we conducted an analysis using ChEMBL (*Gaulton et al., 2012*) and DrugBank databases (*Wishart et al., 2018*). This examination revealed that a total of 11 FDA-approved drugs target the identified genes (*PRKAG2*, *SCN2A*, *DPYD*, *MC3R*, *AKT*, *MAPK*, *KCNK9*, *TACR3*, *HTR4*, and *RXRG*) (*Supplementary file 9*).

## Population validation for associations between age at menarche, or first live birth, or number of births with age-related outcomes

Next, we validated the genetic associations in the population cohort of the UK Biobank. 264,335, 184,481, and 259,758 participants were included for independent variables of age at menarche, age at first live birth, and number of births, respectively (*Supplementary file 10*). Ages at menarche were divided into five age groups, <11, 11–12, 13–14, 15–16, and >16 years. The probability of living to 80 years old, having menopause ≥50 years old, risk of diabetes, HBP, heart failure, COPD, breast

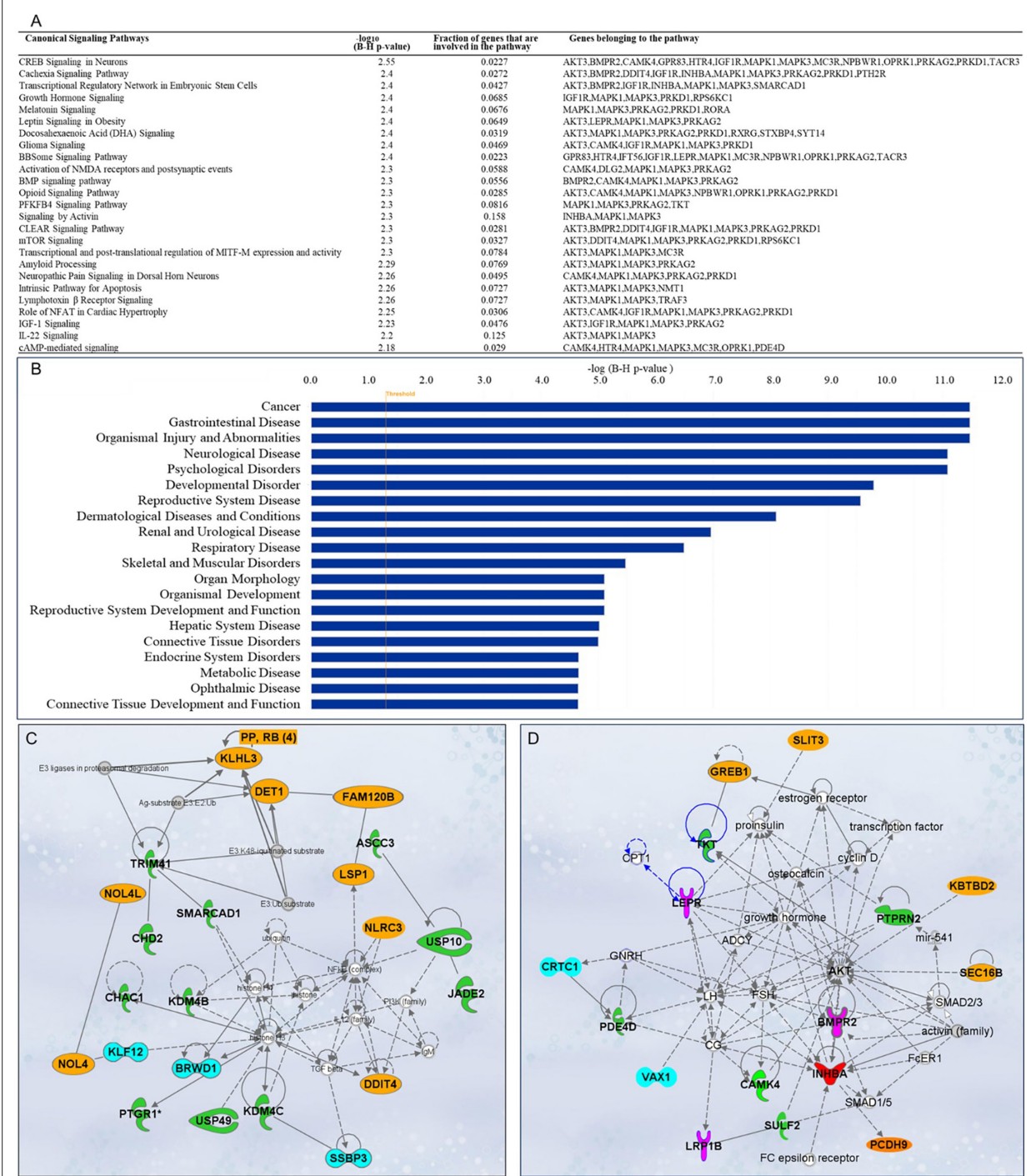

**Figure 5.** Ingenuity Pathway Analysis (IPA). Single-nucleotide polymorphisms (SNPs)/gene outcomes from Mendelian randomization (MR) analysis were subjected to IPA. (**A**) Canonical signaling pathways in age at menarche and first birth. The adjusted p-values of the top 25 signaling pathways are listed. (**B**) Disease and functions in age at first birth and menarche. The adjusted p-value of the top 20 significantly involved diseases and functions is listed. (**C**) IPA network 1 and (**D**) IPA network 2. The genes from our post-MR analysis are in bold. Solid lines indicate direct connections, while dotted lines indicate indirect connections (circular arrows mean influence itself). Color coding: pink—receptors, green—enzymes, blue—transcriptional regulators, red—growth factor, orange—other proteins.

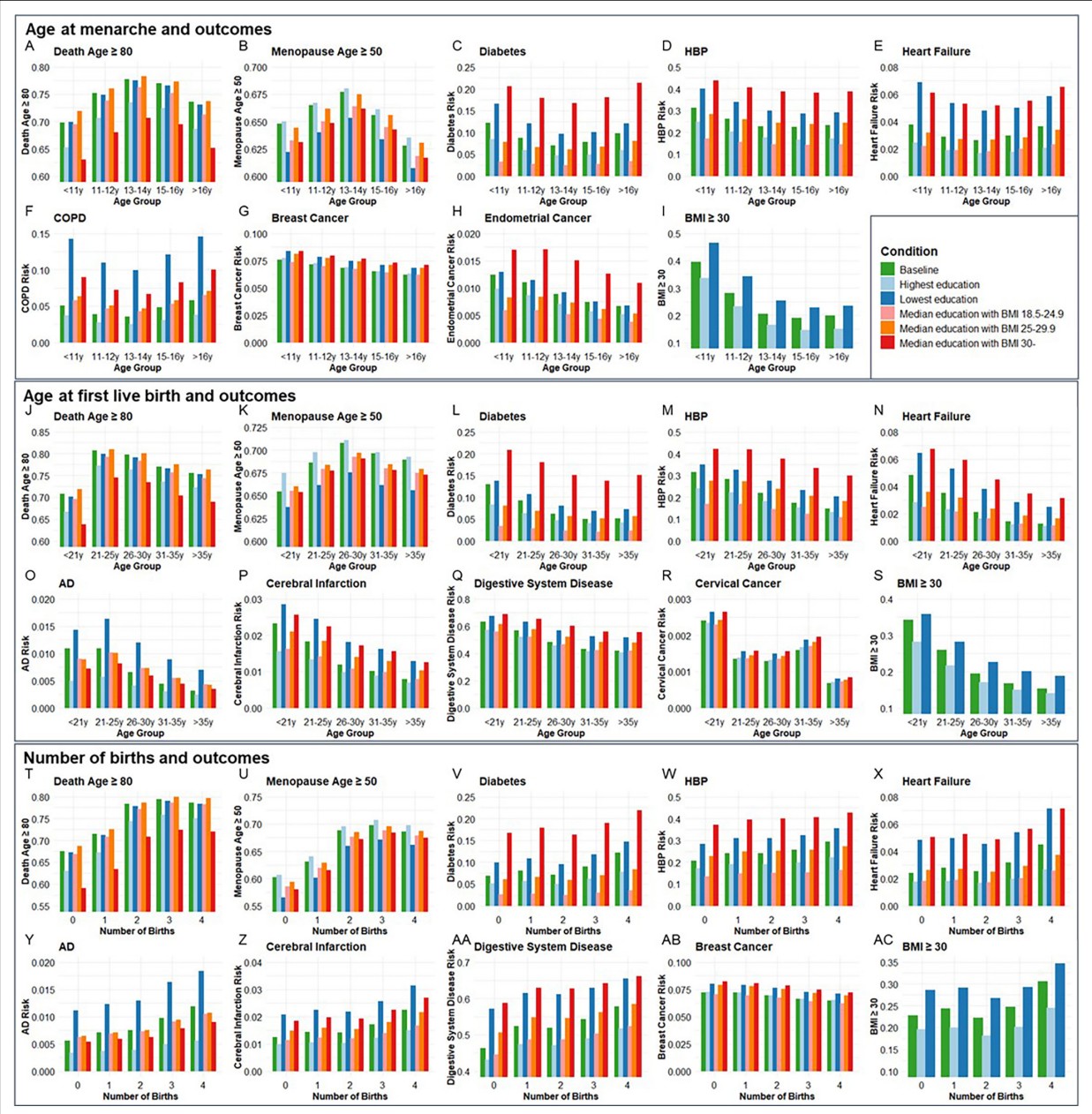

**Figure 6.** The distributions of outcome age and risk according to age group of menarche and first live birth as well as number of births before and after correcting confounders. Condition Baseline, no confounder was corrected. Conditions Highest education and Lowest education, confounders of education, smoking, and drinking were corrected, and bars were painted based on most smoking and drinking situations with the highest and lowest education levels. Conditions Median education with BMI 18.5–24.9, Median education with BMI 25–29.9, and Median education with BMI 30-, confounders of education, smoking, drinking, and BMI were corrected, and bars were painted based on median education and most smoking and drinking situation with three BMI categories. **A-I**, Age at menarche and aging outcomes at different conditions (264,335 participants included). **J-S**, Age at first live birth and aging outcomes at different conditions (184,481 participants included). **T-AC**, Number of births and aging outcomes at different conditions (259,758 participants included). BMI, body mass index; HBP, high blood pressure; COPD, chronic obstructive pulmonary disease; AD, Alzheimer's disease.

The online version of this article includes the following figure supplement(s) for figure 6:

**Figure supplement 1.** The combined effect of age at menarche and first live birth for different outcomes.

cancer, endometrial cancer, and obesity was compared among the five age groups of menarche. Each of the outcomes was compared between the highest and lowest education levels and three BMI categories (18.5–24.9, 25–29.9, and ≥30) among the five age groups (*Figure 6A–I*). The ORs for each age group were calculated based on the coefficients (β) in logistic regression after including confounders of smoking, alcohol, education, and BMI. Compared to the group with menarche under 11 years old, which generally had the highest risk, females with menarche at 13–14 years had the highest probability of living to 80 years old (OR = 1.412, p < 0.001) and having menopause ≥50 years old (OR = 1.146, p < 0.001), and had the lowest risk of diabetes (OR = 0.773, p < 0.001), heart failure (OR = 0.835, p < 0.001), and COPD (OR = 0.728, p < 0.001). The group with menarche at 15–16 years had the lowest risks of HBP (OR = 0.793, p < 0.001) and obesity (OR = 0.337, p < 0.001). Females with menarche >16 years had the lowest risks of breast cancer (OR = 0.832, p < 0.05) and endometrial cancer (OR = 0.641, p < 0.05) compared to females with menarche <11 years (*Supplementary file 11*).

Ages at first birth were divided into five age groups, <21, 21–25, 26–30, 31–35, and >35 years. The probability of living to 80 years old, having menopause ≥50 years old, risk of diabetes, AD, HBP, heart failure, cerebral infarction, digestive system disease, cervical cancer, and obesity was compared among the five age groups of first live birth. Compared to the group with first live birth under 21 years old, which generally had the highest risks, females with first live birth at 21–25 years had the highest probability to live to 80 years old (OR = 1.667, p < 0.001), and females with first live birth at 26–30 years had the highest probability to have menopause ≥50 years old (OR = 1.184, p < 0.001) and lowest risk of cervical cancer (OR = 0.587, p < 0.01). Females with first live birth at 31–35 years had the lowest risk of diabetes (OR = 0.612, p < 0.001). Females with first live birth >35 years had the lowest risk of having AD (OR = 0.474, p < 0.01), HBP (OR = 0.587, p < 0.001), heart failure (OR = 0.446, p < 0.001), cerebral infarction (OR = 0.485, p < 0.001), digestive system disease (OR = 0.569, p < 0.001), and obesity (OR = 0.416, p < 0.001) (*Figure 6J–S*).

Females with the number of births from 0 to 4 were included. Compared to no birth, females with three births had the highest probability of living to 80 years old (OR = 1.818, p < 0.001) and having menopause ≥50 years old (OR = 1.562, p < 0.001). Females with four births had the highest risk of diabetes (OR = 1.399, p < 0.001), AD (OR = 1.676, p < 0.001), HBP (OR = 1.274, p < 0.001), heart failure (OR = 1.439, p < 0.001), cerebral infarction (OR = 1.470, p < 0.001), digestive system disease (OR = 1.370, p < 0.001), and obesity (OR = 1.327, p < 0.001) but lowest risk of breast cancer (OR = 0.869, p < 0.001) (*Figure 6T–AC*). The significant estimated effect (*β*) of each group compared to the youngest age groups of menarche and first live birth and 0 birth group and the ability to explain the variability of outcomes in regression analysis are exhibited in *Supplementary file 11*.

To estimate the combined effect of age at menarche and first live birth on diabetes, HBP, heart failure, and BMI ≥30, participants were divided into 25 groups according to the menarche and first live birth ages. Females with menarche <11 years and first live birth <21 years had higher risks of diabetes (OR = 1.881, p < 0.001), HBP (OR = 1.436, p < 0.05), heart failure (OR = 1.776, p < 0.01), and BMI ≥30 (OR = 4.417, p < 0.001) compared to females with menarche of 13–14 years and first live birth of 26–30 years (*Supplementary file 12* and *Figure 6—figure supplement 1*).

## Discussion

The timing of reproductive milestones such as the age of menarche and age at first birth is partly governed by genetic factors, which may exert lasting impacts on an individual's health trajectory. Our study finds support for antagonistic pleiotropy by demonstrating that genetic variants that drive early menarche or early pregnancy accelerate several aging outcomes, including parental age, frailty index, and several age-related diseases such as diabetes and dementia. These results were validated in a cohort from the UK Biobank, showing that women with early menarche or early pregnancy were associated with accelerated aging outcomes and increased risk of several age-related diseases. These results are also consistent with the disposable soma theory that suggests aging as an outcome trade-off between an organism's investment in reproduction and somatic maintenance and repair. Both antagonistic pleiotropy theory and the disposable soma theory provide valuable evolutionary frameworks for interpreting how reproductive traits influence long-term health, yet they emphasize distinct biological mechanisms. Antagonistic pleiotropy (*Byars and Voskarides, 2020*) highlights the temporal trade-off of gene function, proposing that alleles promoting fertility or reproductive success early in life may

have detrimental effects in later life. In contrast, the disposable soma theory (*van den Heuvel et al., 2016*; *Douglas and Dillin, 2014*) emphasizes a resource allocation trade-off, where investment in reproduction occurs at the cost of somatic maintenance and repair, ultimately accelerating aging and increasing susceptibility to disease. Future research could aim to explore these mechanisms by integrating genetic, metabolic, and longitudinal data, which help understand the interaction between gene-driven effects and resource-driven physiological trade-offs.

Consistent with a recent study suggesting the relation of early first birth with a higher likelihood of frailty (*Guo et al., 2024*), our MR analysis provided a strong association between early age of first birth and GrimAge, a DNA methylation-based marker to predict epigenetic age. Our findings align with the observed negative genetic correlation between reproductive traits and lifespan that individuals with higher polygenic scores for reproduction have lower survivorships to age 76 (*Long and Zhang, 2023*). Prior studies linking the female reproductive factors with aging are either limited to results obtained from observational studies (*Liu et al., 2022*; *Chen et al., 2021*) or are limited to few outcomes such as brain disorders (*Barth and de Lange, 2020*; *Sherwin and Henry, 2008*). Besides, emerging evidence suggests that early puberty in males is linked to adverse health outcomes, such as an increased risk of cardiovascular disease, type 2 diabetes, and hypertension in later life (*Day et al., 2015*). An MR study also reported a genetic association between the timing of male puberty and reduced lifespan (*Hollis et al., 2020*). These findings support the hypothesis that genetic variants associated with delayed reproductive timing in males might similarly confer health benefits or improved longevity, akin to the patterns observed in females. This would suggest that similar mechanisms of antagonistic pleiotropy could operate in males as well.

Using MR, we identified 158 SNPs associated with early menarche or first birth that significantly influence age-related outcomes. Among the top 25 enriched canonical signaling pathways, those involved in developmental and cellular processes are particularly significant in determining the timing of menarche and first childbirth, with nearly one third of these pathways (8 out of 25) falling into this category. These developmental and cellular signaling pathways work together to regulate continuous growth and involution from puberty through menopause (*Jorge et al., 2014*). Interestingly, known longevity pathways such as IGF-1, growth hormone signaling, melatonin signaling, and BMP signaling seem to play a crucial role in regulating the timing of these reproductive events. IGF-1 signaling promotes the growth and development of reproductive organs and tissues (*Baumgarten et al., 2017*; *Bøtkjær et al., 2024*; *Stubbs et al., 2013*), but is also a conserved modulator of longevity (*Papaconstantinou, 2009*; *Kalman, 1969*; *van Heemst, 2010*; *Junnila et al., 2013*). Melatonin, known for regulating circadian rhythms, influences the timing of menarche through its effects on the hypothalamic–pituitary–gonadal axis (*Aleandri et al., 1997*) and is postulated to modulate oxidative, inflammatory, and autophagy states (*Olcese, 2020*). Furthermore, different processes in early life such as cell proliferation or differentiation (*Chen et al., 2004*), ovarian function, and follicular development (*Magro-Lopez and Muñoz-Fernández, 2021*; *Childs et al., 2010*; *Cunha da de et al., 2017*) are dependent on BMP signaling. However, increased BMP signaling also contributes significantly to AD pathology (*Zhang et al., 2021*) and impairments in neurogenesis and cognitive decline associated with aging (*Meyers et al., 2016*). Aberrant BMP signaling is also shown to be associated with age-related metabolic and cardiovascular diseases (*Ye et al., 2023*; *Baboota et al., 2021*). Additionally, growth hormone signaling is not only vital for development during childhood and puberty (*Saenger, 2003*) but also impacts longevity (*Bartke, 2021*). Besides signaling pathways, genes in age at menarche and first childbirth were connected in gene–gene interaction networks. IPA network 1 revealed strong connections of our input genes to regulators involved in chromatin remodeling, immune signaling, and proteostasis. Notably, NF-κB, TGF-β, and IL-12—central players in inflammation and immune modulation—emerged as key hubs. While tightly regulated during early life to support immunity and tissue repair, chronic activation of these pathways is associated with inflammaging and immune dysfunction in later life (*Fülöp et al., 2013*; *Franceschi and Campisi, 2014*). Similarly, links to histone H3/H4 and modifiers like USP10 and USP49 suggest that dysregulation of chromatin and ubiquitin-mediated protein turnover may underlie both developmental gene expression programs and age-related epigenetic drift and proteotoxic stress (*Vilchez et al., 2014*; *Benayoun et al., 2015*). The enrichment of E3 ligase-associated nodes (e.g., E3:K48-ubiquitinated substrate, ubiquitin) further supports a central role for proteostasis networks in aging (*Vilchez et al., 2014*; *Ciechanover and Kwon, 2015*). These associations reflect the principle of antagonistic pleiotropy—where genes and

pathways essential for early-life growth, stress resilience, and immune function become liabilities later in life, contributing to aging and age-related diseases (*Austad and Hoffman, 2018*). In the IPA network 2, most genes were connected with either *AKT* or genes encoding for follicle-stimulating hormone (*FSH*), luteinizing hormone (*LH*), leptin receptor (*LEPR*), and inhibin subunit beta A (*INHBA*). AKT signaling is implicated in oocyte maturation and embryonic development (*Kalous et al., 2023*) and mediates pregnancy-induced cardiac adaptive responses (*Chung et al., 2012*). AKT signaling also mediates age-related disease pathologies, such as AD (*Chen et al., 2019*). Major ovarian functions are controlled by FSH (follicular growth, cellular proliferation, and estrogen production) and LH (oocyte maturation, ovulation, and terminal differentiation of follicles), which in turn are modulated by other ovarian factors such as INHBA (a subunit of activin and inhibin) (*Howard, 2021*; *Bhartiya et al., 2012*). Endocrine alterations at advanced aging or reproductive aging (*Danilovich et al., 2002*) are marked by changes in FSH and LH due to altered feedback resulting from the ovarian decline in sex steroids, inhibin A, and inhibin B production (*Hale et al., 2007*; *Santoro and Randolph, 2011*; *Vanden Brink et al., 2015*). These studies underscore the importance of considering that dysregulation or abnormal activation of any of these pathways could contribute to the onset of late-life diseases consistent with antagonistic pleiotropy in humans (*Byars and Voskarides, 2020*; *Abdellatif et al., 2022*).

The thrifty gene hypothesis provides a compelling framework to explain antagonistic pleiotropy, particularly in the context of modern health challenges. The thrifty gene hypothesis suggests that genes favoring efficient energy storage were advantageous in historical periods of food scarcity, helping individuals survive through famines. However, in today's environment of abundant food, the same genes contribute to obesity and metabolic diseases like diabetes. Similarly, in the context of antagonistic pleiotropy, we hypothesize that genes that enhance early-life reproductive success favor efficient energy storage to support reproductive health. This is supported by our observation that early pregnancy is associated with increasing BMI, a recognized key factor in systemic aging (*Lundgren et al., 2022*; *Quach et al., 2017*). In support of this, we found a lack of significant association between early menarche or age at first birth for CHF and cervical cancer after removal of BMI as a confounder in MR analysis. Based on regression analysis at the population level (*Figure 4*), higher BMI was associated with higher risks of cardiovascular diseases and diabetes. Thus, an increase in BMI is one of the factors that explains the increased risk of age-related disease seen due to the exposure of early pregnancy or the prevalence of genes that enhance early reproductive success. These results are also consistent with calorie restriction being a robust way to extend healthspan and lifespan in many species (*Wilson et al., 2021*).

Several limitations of our study need to be considered. First, all the associations are explored at the genetic level with MR and population level with the UK Biobank. However, to confirm causal relationships, further validation through in vitro and in vivo research is essential. Second, though we excluded confounder-related SNPs and addressed potential pleiotropic effects, there may also be potential bias which could be addressed by extending these findings to other populations. Third, significant heterogeneity and pleiotropy remained in some association analyses, which need further exploration. Fourth, research regarding male-specific reproductive traits and their relationship to aging and health outcomes is needed to compare the difference between male and female aging. Given the increasing age of first childbirth in modern times, the ideal period of first childbirth for both slowing down female aging and benefiting fetal development needs more research (*Balasch, 2010*). In summary, our study underscores the complex relationship between genetic legacies and modern diseases. Understanding these genetic predispositions can inform public health strategies and their relevance to age-related disease outcomes in women of various ethnic groups. For example, interventions could be tailored not only to mitigate the risks associated with early reproductive timing but also to address lifestyle factors exacerbating conditions linked to thrifty genes, like diet and physical activity.

## Acknowledgements

We thank the Kapahi Lab at Buck Institute for helpful discussions. This research has been conducted using the UK Biobank Resource under Application Number 151101.

# Additional information

## Competing interests
Pankaj Kapahi: Reviewing editor, eLife. The other authors declare that no competing interests exist.

## Funding

| Funder | Grant reference number | Author |
| --- | --- | --- |
| Hevolution Foundation | | Pankaj Kapahi |
| National Institute on Aging | R01AG068288 | Pankaj Kapahi |
| National Institute on Aging | R01AG045835 | Pankaj Kapahi |
| Larry L. Hillblom Foundation | | Parminder Singh Pankaj Kapahi |
| Glenn Foundation | | Vikram Pratap Narayan |

The funders had no role in study design, data collection, and interpretation, or the decision to submit the work for publication.

## Author contributions
Yifan Xiang, Vineeta Tanwar, Conceptualization, Resources, Software, Formal analysis, Validation, Investigation, Visualization, Methodology, Writing – original draft, Project administration, Writing – review and editing; Parminder Singh, Conceptualization, Resources, Funding acquisition, Investigation, Visualization, Writing – original draft, Writing – review and editing; Lizellen La Follette, Conceptualization, Methodology, Writing – review and editing; Vikram Pratap Narayan, Pankaj Kapahi, Conceptualization, Resources, Supervision, Funding acquisition, Validation, Investigation, Visualization, Methodology, Writing – original draft, Project administration, Writing – review and editing

## Author ORCIDs
Yifan Xiang http://orcid.org/0000-0002-5559-3654
Parminder Singh https://orcid.org/0000-0002-2851-6805
Pankaj Kapahi https://orcid.org/0000-0002-5629-4947

Reviewer #1 (Public review): https://doi.org/10.7554/eLife.102447.4.sa1
Reviewer #2 (Public review): https://doi.org/10.7554/eLife.102447.4.sa2
Author response https://doi.org/10.7554/eLife.102447.4.sa3

---

# Additional files

## Supplementary files
MDAR checklist

Supplementary file 1. Overview of GWAS data used for MR analysis.

Supplementary file 2. SNPs included in MR analysis for each outcome.

Supplementary file 3. Overview results of two-sample MR analysis.

Supplementary file 4. Overview results of two-step MR analysis with BMI as the mediator.

Supplementary file 5. Results of two-sample MR analysis after colocalization.

Supplementary file 6. The genetic correlation based on LDSC.

Supplementary file 7. List of SNPs/genes from post-MR analysis showing an association between age at menarche onset and first birth with one or two aging outcomes.

Supplementary file 8. Ingenuity pathway analysis networks in age at menarche onset and first birth.

Supplementary file 9. FDA-approved drugs against targets identified post-MR analysis.

Supplementary file 10. Participants included for population validation from UK Biobank.

Supplementary file 11. Estimate effect compared to youngest/smallest group for outcomes in regression.

Supplementary file 12. Estimate effect of combined age at menarche and first live birth and sample distribution.

## Data availability

All data are available in the main text, supplementary materials, or public resources. The population data is available through the UK Biobank.

The following previously published datasets were used:

| Author(s) | Year | Dataset title | Dataset URL | Database and Identifier |
|---|---|---|---|---|
| Loh PR | 2018 | Menarche (age at onset) | https://gwas.mrcieu.ac.uk/datasets/ebi-a-GCST90029036/ | OpenGWAS, ebi-a-GCST90029036 |
| Mills MC | 2021 | Age at first birth | https://gwas.mrcieu.ac.uk/datasets/ebi-a-GCST90000048/ | OpenGWAS, ebi-a-GCST90000048 |
| Atkins JL | 2021 | Frailty index | https://gwas.mrcieu.ac.uk/datasets/ebi-a-GCST90020053/ | OpenGWAS, ebi-a-GCST90020053 |
| Pilling LC | 2017 | Parental longevity (father's age at death) | https://gwas.mrcieu.ac.uk/datasets/ebi-a-GCST006700/ | OpenGWAS, ebi-a-GCST006700 |
| Pilling LC | 2017 | Parental longevity (mother's age at death) | https://gwas.mrcieu.ac.uk/datasets/ebi-a-GCST006699/ | OpenGWAS, ebi-a-GCST006699 |
| McCartney DL | 2021 | DNA methylation GrimAge acceleration | https://gwas.mrcieu.ac.uk/datasets/ebi-a-GCST90014294/ | OpenGWAS, ebi-a-GCST90014294 |
| Loh PR | 2018 | Menopause (age at onset) | https://gwas.mrcieu.ac.uk/datasets/ebi-a-GCST90029037/ | OpenGWAS, ebi-a-GCST90029037 |
| Kunkle B | 2019 | IGAP Rare Variants: Stage 1 (GRCh38) | https://www.niagads.org/genomics/app/record/track/NG00075_GRCh38_STAGE1 | NIAGADS Alzheimer's GenomicsDB, NG00075_GRCh38_STAGE1 |
| Dönertaș HM | 2021 | Osteoporosis | https://gwas.mrcieu.ac.uk/datasets/ebi-a-GCST90038656/ | OpenGWAS, ebi-a-GCST90038656 |
| Saori S | 2021 | Type 2 diabetes | https://gwas.mrcieu.ac.uk/datasets/ebi-a-GCST90018926/ | OpenGWAS, ebi-a-GCST90018926 |
| Sakaue S | 2021 | Chronic heart failure | https://gwas.mrcieu.ac.uk/datasets/ebi-a-GCST90018806/ | OpenGWAS, ebi-a-GCST90018806 |
| Elsworth B | 2018 | Diagnoses - secondary ICD10: I10 Essential (primary) hypertension | https://gwas.mrcieu.ac.uk/datasets/ukb-b-12493/ | OpenGWAS, ukb-b-12493 |
| Elsworth B | 2018 | Facial ageing | https://gwas.mrcieu.ac.uk/datasets/ukb-b-2148/ | OpenGWAS, ukb-b-2148 |
| Wojcik GL | 2019 | Chronic kidney disease | https://gwas.mrcieu.ac.uk/datasets/ebi-a-GCST008026/ | OpenGWAS, ebi-a-GCST008026 |
| MRC Integrative Epidemiology Unit at the University of Bristol | 2021 | Early onset COPD | https://gwas.mrcieu.ac.uk/datasets/finn-b-COPD_EARLY/ | OpenGWAS, finn-b-COPD_EARLY |

*Continued*

| Author(s) | Year | Dataset title | Dataset URL | Database and Identifier |
|-----------|------|---------------|-------------|-------------------------|
| Dönertaş HM | 2021 | Gastrointestinal or abdominal disease | https://gwas.mrcieu.ac.uk/datasets/ebi-a-GCST90038597/ | OpenGWAS, ebi-a-GCST90038597 |
| Sakaue S | 2021 | Breast cancer | https://gwas.mrcieu.ac.uk/datasets/ebi-a-GCST90018799/ | OpenGWAS, ebi-a-GCST90018799 |
| Sakaue S | 2021 | Ovarian cancer | https://gwas.mrcieu.ac.uk/datasets/ebi-a-GCST90018888/ | OpenGWAS, ebi-a-GCST90018888 |
| Elsworth B | 2018 | Cancer code, self-reported: cervical cancer | https://gwas.mrcieu.ac.uk/datasets/ukb-b-8777/ | Elsworth, ukb-b-8777 |
| Elsworth B | 2018 | Body mass index | https://gwas.mrcieu.ac.uk/datasets/ukb-b-19953/ | OpenGWAS, ukb-b-19953 |
| Kunkle B | 2019 | IGAP Rare Variants: Stage 2 (GRCh38) | https://www.niagads.org/genomics/app/record/track/NG00075_GRCh38_STAGE2 | NIAGADS Alzheimer's GenomicsDB, NG00075_GRCh38_STAGE2 |
| Tracy AOM | 2018 | Endometrial cancer | https://gwas.mrcieu.ac.uk/datasets/ebi-a-GCST006464/ | OpenGWAS, ebi-a-GCST006464 |
| Emdin CA | 2021 | Cirrhosis, broad definition used in the article | https://gwas.mrcieu.ac.uk/datasets/finn-b-CIRRHOSIS_BROAD/ | OpenGWAS, finn-b-CIRRHOSIS_BROAD |

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
