## [Editor Report · eLife Assessment]

This **important** study uses Mendelian Randomization to provide evidence that early-life reproductive phenotypes (i.e., age at onset of menarche and age at first birth) have a significant impact on numerous health outcomes later in life. The empirical evidence provided by the authors supporting the antagonistic pleiotropy theory is **solid**. Theories of aging should be empirically tested, and this study provides a good first step in that direction.

---

## [Referee Report · Reviewer #1 (Public review)]

Summary:

The present study aims to determine possible associations between reproduction with prevalence of age-related diseases based on the antagonistic pleiotropy hypothesis of ageing predominantly using Mendelian Randomization. The authors provide evidence demonstrated that menarche before the age 11 and childbirth before 21 increases the risk of several diseases, and almost doubled the risk for diabetes, heart failure, and quadrupled the risk of obesity,

Strengths:

Large sample size. Many analyses

---

## [Referee Report · Reviewer #2 (Public review)]

Summary:

The authors present an interesting paper where they test the antagonistic pleiotropy theory. Based on this theory they hypothesize that genetic variants associated with later onset of age at menarche and age at first birth have a positive causal effect on a multitude of health outcomes later in life, such as epigenetic aging and prevalence of chronic diseases. Using a mendelian randomization and colocalization approach, the authors show that SNPs associated with later age at menarche are associated with delayed aging measurements, such as slower epigenetic aging and reduced facial aging and a lower risk of chronic diseases, such as type 2 diabetes and hypertension. Moreover, they identify 128 fertility-related SNPs that associate with age-related outcomes and they identified BMI as a mediating factor for disease risk, discussing this finding in the context of evolutionary theory.

Strengths:

The major strength of this manuscript is that it addresses the antagonistic pleiotropy theory in aging. Aging theories are not frequently empirically tested although this is highly necessary. The work is therefore relevant for the aging field as well as beyond this field, as the antagonistic pleiotropy theory addresses the link between fitness (early life health and reproduction) and aging.

Weaknesses:

The authors report evidence in support of the antagonistic pleiotropy theory in aging and discuss the discuss the disposable soma theory. Although both theories describe distinct mechanisms, separating them in empirical research is complicated and needs further studies in future research.

---

## [Author Response]

The following is the authors’ response to the previous reviews

**Reviewer #1 (Public review):**
Summary:The present study aims to associate reproduction with age-related disease as support of the antagonistic pleiotropy hypothesis of ageing predominantly using Mendelian Randomization. The authors found evidence that early-life reproductive success is associated with advanced ageing.Strengths:Large sample size. Many analyses.Weaknesses:Still a number of doubts with regard to some of the results and their interpretation.
**Reviewer #1 (Recommendations for the authors):**
Thank you for the opportunity to review a revised version.I still have serious doubts with regard to a number of datasets presented. For example, the results on essential hypertension and cervical cancer show very small effect sizes, but according to the authors still reach the level of statistical significance. This is unlikely to be accurate. For MR analyses, this is nearly impossible. The analyses of these data and the statistical analysis need to be checked for errors and repeated. While BOLT-LLM might not be relevant here, there might be other things happening here. The authors should therefore always interpret the results also with regard to the observed effect sizes instead of only looking at the p-values (0.999 means that there is a 0.1% lower risk).

Thank you for your suggestions. We have updated the results for essential hypertension, GAD, and cervical cancer in results, figures, and supplemental tables (lines 65-89, Figure 1, Tables S3-S4).

**Reviewer #2 (Public review):**
Summary:The authors present an interesting paper where they test the antagonistic pleiotropy theory. Based on this theory they hypothesize that genetic variants associated with later onset of age at menarche and age at first birth may have a positive effect on a multitude of health outcomes later in life, such as epigenetic aging and prevalence of chronic diseases. Using a mendelian randomization and colocalization approach, the authors show that SNPs associated with later age at menarche are associated with delayed aging measurements, such as slower epigenetic aging and reduced facial aging and a lower risk of chronic diseases, such as type 2 diabetes and hypertension. Moreover, they identify 128 fertility-related SNPs that associate with age-related outcomes and they identified BMI as a mediating factor for disease risk, discussing this finding in the context of evolutionary theory.Strengths:The major strength of this manuscript is that it addresses the antagonistic pleiotropy theory in aging. Aging theories are not frequently empirically tested although this is highly necessary. The work is therefore relevant for the aging field as well as beyond this field, as the antagonistic pleiotropy theory addresses the link between fitness (early life health and reproduction) and aging.The authors addressed the remarks on the previous version very well. Addressing the two points below would further increase the quality of the manuscript.(1) In the previous version the authors mentioned that their results are also consistent with the disposable soma theory: "These results are also consistent with the disposable soma theory that suggests aging as an outcome tradeoff between an organism's investment in reproduction and somatic maintenance and repair."Although the antagonistic pleiotropy and disposable soma theories describe different mechanisms, both provide frameworks for understanding how genes linked to fertility influence health. The antagonistic pleiotropy theory posits that genes enhancing fertility early in life may have detrimental effects later. In contrast, the disposable soma theory suggests that energy allocation involves a trade-off, where investment in fertility comes at the expense of somatic maintenance, potentially leading to poorer health in later life.To strengthen the manuscript, a discussion section should be added to clarify the overlap and distinctions between these two evolutionary theories and suggest directions for future research in disentangling their specific mechanisms.

Thank you for your suggestions to clarify the overlap and distinctions between the antagonistic pleiotropy and disposable soma theories. While our primary focus is on the antagonistic pleiotropy framework, we acknowledge that the disposable soma theory also provides a relevant perspective on the trade-offs between reproduction and somatic maintenance.

To address this, we have expanded the discussion section to highlight how both theories contribute to our understanding of the relationship between fertility-related traits and aging-related health outcomes. We also suggested potential future research directions, such as integrating genetic data with biomarkers of somatic to further explore the mechanisms underlying these trade-offs (lines 213-223).

(2) In response to the question why the authors did not include age at menopause in addition to the already included age at first child and age at menarche the following explanation was provided: "Our manuscript focuses on the antagonistic pleiotropy theory, which posits that inherent trade-off in natural selection, where genes beneficial for early survival and reproduction (like menarche and childbirth) may have costly consequences later. So, we only included age at menarche and age at first childbirth as exposures in our research."It remains, however, unclear why genes beneficial for early survival and reproduction would be reflected only in age at menarche and age at first childbirth, but not in age at menopause. While age at menarche marks the onset of fertility, age at menopause signifies its end. Since evolutionary selection acts directly until reproduction is no longer possible (though indirect evolutionary pressures persist beyond this point), the inclusion of additional fertility-related measures could have strengthened the analysis. A more detailed justification for focusing exclusively on age at menarche and first childbirth would enhance the clarity and rigor of the manuscript.

Thank you for your question regarding the age at menopause in our analysis. Our decision was based on the theoretical framework of antagonistic pleiotropy, which emphasizes early-life reproductive advantages that may have trade-offs later in life. Age at menarche and age at first childbirth are direct markers of early reproductive investment, which align closely with this framework.

While age at menopause marks the cessation of reproductive capability, its evolutionary role is distinct. The selective pressures acting on menopause are complex and may involve post-reproductive contributions rather than direct reproductive fitness benefits. Moreover, the genetic architecture of menopause may be influenced by different biological pathways compared to early reproductive traits.

Nonetheless, we acknowledge that including age at menopause could provide additional insights into reproductive aging. Several papers1,2 were already published regarding age at menopause and age-related outcomes, including diabetes, AD, osteoporosis, cancers, and cardiovascular diseases.

**Reviewing Editor (Recommendations for the authors):**
Above/below you will find the remaining comments from the reviewers. One of the main issues remaining is that some of the data seems to be incorrectly analysed and some of the findings may not be correct. To clarify this a lot more, I asked the reviewer for some details and received the following:- In Figure 1B one of their main outcomes is "age of menopause", but they report the data as an odds ratio. This is not correct and should be fixed (it seems the authors can run the right analysis, but just reported it with the wrong heading in the figure). This likely also applies to the outcome "facial aging". Also the heading in Figure 1A should be Beta instead of OR.

We have updated the figures to ensure that the beta values of continuous outcomes and odds ratio values of categorical outcomes are presented in Figure 1.

- With essential hypertension, GAD and cervical cancer, the estimates are so small that they need to re-review their results. The current MR analysis is not sufficiently powered to have such small confidence intervals. Essential hypertension was based on data from UK biobank, although I was also unable to find what program was used to generate the GWAS results, I have strong thoughts this was also BOLT-LLM. Same for cervical cancer. Both datasets used familial-related samples, so they are very likely derived with BOLT-LLM.I hope this will help to solve this issue.

Based on published paper, gastrointestinal or abdominal disease (GAD) (GWAS ID: ebi-a-GCST90038597) is after BOLT-LLM. Based on MRC IEU UK Biobank GWAS pipeline, version 1 and 2, essential hypertension (GWAS ID: ukb-b-12493) and cervical cancer (GWAS ID: ukb-b-8777) are after BOLT-LLM. We have updated the MR analysis results and figures (lines 65-89, Figure 1, Tables S3-S4) as well as the following IPA analysis (lines 106-162 and 255-280, Figures 2-3).

(1) Magnus, M. C., Borges, M. C., Fraser, A. & Lawlor, D. A. Identifying potential causal effects of age at menopause: a Mendelian randomization phenome-wide association study. Eur J Epidemiol 37, 971-982 (2022). https://doi.org:10.1007/s10654-022-00903-3

(2) Zhang, X., Huangfu, Z. & Wang, S. Review of mendelian randomization studies on age at natural menopause. Front Endocrinol (Lausanne) 14, 1234324 (2023). https://doi.org:10.3389/fendo.2023.1234324